# The Role of CaMKII and ERK Signaling in Addiction

**DOI:** 10.3390/ijms22063189

**Published:** 2021-03-20

**Authors:** Wenbin Jia, Ichiro Kawahata, An Cheng, Kohji Fukunaga

**Affiliations:** Department of Pharmacology, Graduate School of Pharmaceutical Sciences, Tohoku University, Sendai 980-8578, Japan; jia.wenbin.t6@dc.tohoku.ac.jp (W.J.); kawahata@tohoku.ac.jp (I.K.); cheng.an.q6@dc.tohoku.ac.jp (A.C.)

**Keywords:** nicotine-induced addiction, Ca^2+^/calmodulin-dependent protein kinase II, extracellular signal-regulated kinase, dopamine D1 receptor, dopamine D2 receptor, fatty acid-binding protein 3

## Abstract

Nicotine is the predominant addictive compound of tobacco and causes the acquisition of dependence through its interactions with nicotinic acetylcholine receptors and various neurotransmitter releases in the central nervous system. The Ca^2+^/calmodulin-dependent protein kinase II (CaMKII) and extracellular signal-regulated kinase (ERK) play a pivotal role in synaptic plasticity in the hippocampus. CaMKII is involved in long-term potentiation induction, which underlies the consolidation of learning and memory; however, the roles of CaMKII in nicotine and other psychostimulant-induced addiction still require further investigation. This article reviews the molecular mechanisms and crucial roles of CaMKII and ERK in nicotine and other stimulant drug-induced addiction. We also discuss dopamine (DA) receptor signaling involved in nicotine-induced addiction in the brain reward circuitry. In the last section, we introduce the association of polyunsaturated fatty acids and cellular chaperones of fatty acid-binding protein 3 in the context of nicotine-induced addiction in the mouse nucleus accumbens and provide a novel target for the treatment of drug abuse affecting dopaminergic systems.

## 1. Introduction

Smoking was the second leading cause of early death and disability in 2015 [1]. It caused approximately 6.4 million deaths, which accounted for 11.5% of all deaths worldwide, and led to 150 million people becoming disabled in 2015 [2]. In developed countries, inhalation of smoking is considered the largest single cause of early death. Smoking is the reason for 20% of all premature deaths and more than one-third of all deaths in men aged 35–69 years [3]. Accumulated evidence suggests that regular smoking can enhance the risk of coronary heart disease, chronic obstructive pulmonary disease, and cancers of the upper aerodigestive tract and lung [4]. Nicotine in tobacco is considered the predominant addictive component, and it causes continued use in humans [5,6,7]. Nicotine-induced addiction is reported to be difficult to quit. More than 80% of attempts to quit fail within a year, and some highly addicted smokers are able to quit smoking only for a few hours [8,9]. Therefore, people are inclined to use tobacco continually despite the harmful consequences. Other psychostimulant drugs, such as morphine, cocaine, and amphetamine, are similar to nicotine: they participate and modulate the brain reward system and motivation though they have different interaction sites and chemical structures [10].

Ca^2+^/calmodulin-dependent protein kinase II (CaMKII) is involved in synaptic plasticity, long-term potentiation (LTP), and memory consolidation in the hippocampus [11,12]. However, the engagement of CaMKII in the brain reward system and psychostimulants, including nicotine-induced addiction, has not been entirely elucidated. In this review, we focus on the mechanisms of CaMKII and extracellular signal-regulated kinase (ERK), which is mediated by CaMKII function in the brain reward system, in the context of nicotine and other psychostimulant-induced addiction. We also consider the participation of CaMKII in both dopamine D1 receptor (D1R) and dopamine D2 receptor (D2R) signaling, provide fatty acid-binding protein 3 (FABP3) as a novel target, and clarify the underlying mechanisms in terms of nicotine and other psychostimulant-induced addiction affecting the dopaminergic (DAergic) system.

## 2. Mechanisms Involved in Nicotine-Induced Addiction in the Brain

Nicotine in tobacco is inhaled into the lung and is subsequently moved to the brain through arterial circulation. Nicotine initiates its biological function by interacting with nicotinic acetylcholine receptors (nAChRs) through binding with pentameric ligand-gated ion channels, leading to the rotation of the receptors as well as the opening of cation channels [13,14]. Nicotine in tobacco plays a role as an extrinsic agonist of nAChRs, binds at the boundary surface between two subunits of the receptor, and causes the opening of the ion channel, which provides a water-filled pathway through the membrane, and the influx of calcium and sodium [15]. The nAChR is composed of five polypeptide subunits, and a diverse combination of subunits causes the formation of various nAChR subtypes [16]. Nine α (α2–α10) and three β subunits (β2–β4) are mainly expressed in the mammalian brain [17,18]. According to its high affinity to nicotine and abundance in the midbrain dopamine (DA) pathways, α4β2 nAChR is considered the dominant target in nicotine-induced addiction [19,20]. The α4 or β2 subunit null mice exhibit impaired nicotine-seeking behaviors; reinserting these ablated genes into the α4 or β2 null mice ventral tegmental area (VTA) can rescue the dysfunction of behavioral effects in response to nicotine, which suggests that VTA α4β2 nAChRs are crucial for nicotine-induced addiction [21,22,23]. Chronic nicotine administration leads to an increase in the number of α4β2 nAChR binding sites in the rat brain [24]. Some studies reported that the reinsertion of low levels of α4β2 nAChRs could not produce significant conditioned place preference (CPP) in mice, suggesting that the β2 subunit needs a threshold level to fulfill biological functions [25]. The α7 homomeric receptors are extensively expressed in the mammalian central nervous system (CNS) and have higher permeability to calcium as well as faster kinetics than other nAChRs [16,26]. The administration of the α7 nAChR agonist improved learning and working memory in rats [27]. The α7 nAChRs are able to regulate long-term potentiation (LTP) to DAergic neurons at glutamatergic inputs [28,29,30].

The activation of nAChRs promotes various neurotransmitter releases in the CNS [15,31]. Dopamine (DA), a predominant neurotransmitter involved in drug abuse, plays a critical role in reinforcing behaviors in the drug reward system [32,33,34]. The results of microdialysis using freely moving rats suggested that stimulant drugs such as nicotine, cocaine, and amphetamine increase extracellular DA concentrations in the mesolimbic system [35]. DA neuron stimulation is a fundamental characteristic of drug abuse and causes drug dependence [35,36]. Some studies have revealed that the brain has specific regions for reward functions, and electrically stimulating these regions has a high reward since rats made operational responses when these regions were stimulated [37]. The mesolimbic DAergic projections originating from the ventral tegmental area (VTA) and projected to the nucleus accumbens (NAc) and the prefrontal cortex (PFC) are particularly susceptible to electrical stimulation. This system has been represented as a neurochemical substrate in reward functions since it is fully involved in the stimulant drug reward system [38]. The DAergic projections also include the olfactory tubercle, septal region, and amygdala. Many investigations have revealed that DA levels in the NAc are elevated not only by the action of natural rewards, such as water, food, and sex, but also by stimulant drugs, including nicotine, cocaine, and amphetamine, especially in the dependence acquisition phase [6,39,40,41]. Mesolimbic DAergic system lesions produced by the microinfusion of 6-hydroxydopamine into the NAc leads to a significant reduction in nicotine self-administration in rats [42]. The impairment of food-induced DA release in the NAc shell provoked by the microinjection of naloxonazine into the VTA in rats suggests that mesolimbic DAergic projections are critical in DA release and the process of dependence [43]. The NAc is a heterogeneous structure composed of two major subdivisions, the NAc core and NAc shell, which can be considered as a part of the extended amygdala [44]. Although both of these subdivisions play an essential role in learning behaviors and classical conditioning, as they are anatomically different and have distinct projection areas, they are considered to have different contributions [45,46]. The NAc core projects to the extended amygdala, lateral hypothalamus, and central gray matter, and plays a critical role in the response of induced or reinforced conditioned stimulation [45,47,48]. However, the NAc shell is correlated with the acquisition of responding engaged in stimulant drug reinforcement [49]. Some investigations have reported that DA overflow in the medial NAc shell is stimulated by acute nicotine administration, while the extracellular levels of DA in the NAc core were not obviously increased following the same administration [41,50,51]. Intravenously administration with other addictive drugs such as cocaine, morphine, and amphetamine also preferentially enhance extracellular DA levels in the NAc shell relative to the NAc core, demonstrating that the NAc shell has a priority for the effect of psychostimulant drugs at doses that could maintain intravenous drug self-administration [40].

Nicotine stimulation also promotes the release of glutamate (Glu) and γ-aminobutyric acid (GABA), of which the former facilitates and the latter suppresses DA release [28,29]. Some of the nAChR subtypes reach desensitization states following chronic nicotine exposure, while the other receptors do not. In one study, the GABA-mediated inhibitory function was weakened while Glu-mediated excitatory function was still sustained, resulting in the elevation of the excitation of DAergic neurons and enhancement of responsiveness to nicotine [52]. The nAChRs become desensitized under long exposure to nicotine, which seems to be the reason for their increase [53,54]. Nicotine withdrawal causes excessive nAChRs to ameliorate from the state of desensitization and leads to hyperexcitation in the nicotinic cholinergic systems, thereby promoting patients or nicotine-induced dependent rodents to obtain more nicotine. Nicotine relapse contributes to the desensitization of an excessive number of nAChRs to normal levels [16].

## 3. CaMKII in Nicotine and Other Psychostimulant-Induced Addiction

CaMKII plays a critical role in the molecular pathway of the reward system in drug addiction, including nicotine dependence, and affects animal responses to drug abuse [55]. Previous investigations demonstrated that 14 or 28 consecutive days of chronic nicotine administration significantly increased CPP scores in conditioning, nicotine withdrawal, as well as relapse phases and CaMKII autophosphorylation levels in the mouse NAc and hippocampal CA1 region [56]. The number of CaMKII autophosphorylation-positive cells in the NAc was also elevated following chronic nicotine administration relative to saline-administered mice [56]. Intracerebroventricular infusion of CaMKII antagonists KN-62 and KN-93 into mice showed impaired nicotine-induced CPP [57]. CaMKIIα heterozygous (+/−) mice failed to show nicotine-induced CPP behavior, and the increased CaMKII activity in the NAc and VTA in wild-type (WT) mice were blocked by the administration of KN-62 [57]. Nicotine induces significant CPP and elevation of CaMKII activity in a concentration-dependent manner. Some studies identified that intraperitoneal injection of nicotine at doses of 0.25 mg/kg and 1.0 mg/kg could not produce nicotine-induced conditioning, whereas that at a dose of 0.5 mg/kg could [58,59]. The concentration at 2 mg/kg induced conditioned place aversion but not CPP in mice [58]. Nicotine concentration lower than 0.1 mg/kg or higher than 1.4 mg/kg could not induce CPP in subcutaneously injected rats [60]. Although different neurochemical outcomes can be observed with different concentrations of nicotine, our previous data suggest that the daily dose of nicotine at 0.5 mg/kg could execute its functions that induce significant CPP, and elevate CaMKII autophosphorylation level as well as the phosphorylation level of its downstream targets in the NAc and hippocampal CA1 region using an immunoblotting technique [56]. Consistent with nicotine, other addictive drugs such as morphine and methamphetamine have a similar effect with regard to the interaction with CaMKII. CaMKII modulates memory retention in an N-methyl-D-aspartate receptor (NMDAR)-dependent manner in morphine-sensitized rats [61]. Reduced activations of CaMKII and cAMP response element-binding protein (CREB) were observed in the morphine self-administered rat NAc following treatment with a TRPV1 antagonist, which is from the transient receptor potential (TRP) cation channel family [62]. The autophosphorylation level of CaMKII in the limbic forebrain was increased in mice administered with morphine, and this upregulation was blocked by the intracerebroventricular infusion of KN-93 [63]. The CaMKII signal is fully involved in the effect of cocaine- and amphetamine-regulated transcript (CART) peptide, which is a neuropeptide associated with brain reward circuits, in the context of cocaine reward [64]. In the NAc, CaMKIIα contributes to the psychomotor effects induced by cocaine [65], reinstatement of cocaine-seeking behaviors [66], and cocaine-induced CPP behaviors [67,68]. CaMKIIα expressed in other brain regions such as the prefrontal cortex (PFC) and amygdala is also correlated with cocaine-induced CPP and cue-induced cocaine-seeking behaviors [67,69]. Furthermore, CaMKIIα autophosphorylation levels were significantly elevated in the PFC, hippocampus, and ventral striatum in rats following ketamine self-administration [70]. Here, we introduce some evidence suggesting that the activities of CaMKII (Table 1) are fully involved in certain brain regions located in the brain reward circuitry. Exposure to the stimulant drugs does not cause addiction immediately; it involves various neuronal adaptations that develop over time.

These stimulant drugs have been reported to activate and alter the reward circuitry of the brain, and affect synaptic plasticity, learning acquisition, and glutamatergic inputs into the brain neuronal circuit [77,78] (Figure 1). The brain reward circuitry underlying the addiction process is complicated, which mainly includes DA release from DAergic neurons in the VTA of the midbrain and projections to the NAc and PFC region. CaMKIIα autophosphorylation levels were significantly elevated in the PFC of morphine-administered mice [73]. VTA plays a critical role in the initial stimulant drug exposure and causes long-term adaptation in DAergic neurons in the projection regions [79]. Stimulant drug exposure increases DA signaling in the NAc and functionally regulates glutamatergic excitatory projections to the NAc medium spiny neurons (MSNs). Addictive drugs trigger and modulate synaptic plasticity in the brain reward circuitry involved in addiction. Some studies have suggested that alterations in synaptic plasticity at glutamatergic inputs from the cortex to the NAc may be the underlying mechanism in stimulant drug addiction [77,80]. CaMKII is critically associated with the NAc shell DA and glutamatergic inputs involved in synaptic plasticity [66].

## 4. ERK in Nicotine and Other Psychostimulant-Induced Addiction

ERK is a serine-threonine protein kinase and a member of the mitogen-activated protein kinase (MAPK) family. It has been reported to play essential roles in cellular proliferation, differentiation, neuronal survival, and synaptic plasticity [84,85]. ERK has two isoforms, ERK1 (p44 MAPK) and ERK2, which are both expressed throughout the mesolimbic system in the mouse brain, including the regions such as the PFC, NAc, amygdala, and VTA [86]. ERK is recruited for signaling transfer in response to extracellular stimulations and the initiated signal cascade. BDNF activates receptor tyrosine kinases (RTKs), which are correlated with adapter protein recruitment such as Shc, and causes the activation of GTPase Ras, serine-threonine kinase Raf, and MAPK-ERK kinase (MEK), which in turn, phosphorylate and activate ERK. Moreover, Ras is also activated through L-type calcium channels and NMDAR. The Ca^2+^ influx evoked by the electrical and pharmacological activation of these channels and receptors plays a critical role in the signal cascade. Furthermore, dopamine D1 receptors (D1Rs) could also elevate Ca^2+^ influx and lead to the activation of protein kinase A (PKA) and Raf, which further causes ERK phosphorylation [87]. The phosphorylation of ERK leads to the activation of downstream targets such as Elk-1 transcription factor [88], and CREB through mitogen- and stress-stimulated kinase 1 (MSK1) and ribosomal S6 kinase (RSK) [89,90]. Since ERK is regulated by DAergic and glutamatergic signals, it has been suggested that ERK is involved in psychostimulant-induced addiction and plays a crucial role in the brain reward system and learning process [91]. Here, we provide a simplified schematic representation of the ERK signal cascade involved in psychostimulant-induced addiction (Figure 2).

The activation of ERK is mediated by CaMKII in stimulant drug-induced addiction. Nicotine-induced elevation of ERK phosphorylation was found to be blocked in mouse primary cortical neurons following treatment with the CaMKII inhibitor KN-93 [94]. Treatment with KN-93 also significantly reduced nicotine-induced ERK phosphorylation levels in PC12h cells [95]. The calcium chelator BAPTA completely blocked nicotine-induced ERK phosphorylation, and L-type calcium channel blocker significantly decreased ERK phosphorylation levels following nicotine treatment, which suggests that Ca^2+^ is critical in the ERK phosphorylation process [94]. Moreover, the crosslinking and co-immunoprecipitation experiments proved that CaMKII forms a tight complex and interacts with NMDAR in living cells [96,97]. ERK phosphorylation induced by cocaine treatment is in accordance with the NMDAR function that NMDAR activation plays a role in neuronal adaptions involved in drug abuse [98]. Based on the evidence that Raf, the upstream target of ERK, is activated by Ca^2+^ influx through NMDAR, it is reasonable that CaMKII mediates and plays a critical role in ERK phosphorylation, which is involved in psychostimulant-induced addiction. Consistent with CaMKII, the activation of ERK is also involved in nicotine and other drugs of abuse. Some studies revealed that seven consecutive days of nicotine administration successfully elevated ERK phosphorylation levels in the rat NAc [99]. Subcutaneous administration of nicotine into adolescent rats caused a robust increase in ERK1/2 phosphorylation levels in the nucleus accumbens shell [100]. ERK1/2 phosphorylation levels were significantly upregulated following nicotine treatment in PC12h cells, while this elevation was inhibited by treatment with the MEK inhibitor, U0126 [95]. Furthermore, morphine induced significantly increased mice CPP behaviors, and ERK phosphorylation levels in the VTA were inhibited following treatment with the inhibitor, compound 511 [101]. Methamphetamine administration significantly increased the behavioral sensitization evaluated by the total distance in mice, and elevated ERK phosphorylation and ΔFosB levels in the mouse NAc [102]. Methamphetamine injection also elevated the expression of GluN2B, which in turn caused a significant increase in ERK, CREB phosphorylation, and BDNF levels [103]. Ethanol-induced enhancement of ethanol sensitization and ERK phosphorylation levels were observed in the mouse NAc shell, and these elevations were inhibited by pretreatment with MPEP, which is an mGluR5 antagonist [104]. Here, we summarize some evidence revealing that ERK activity is related to nicotine and other psychostimulant-induced addiction in the brain reward circuitry (Table 2). As a consequence of nicotine and other psychostimulants in activation of CaMKII and ERK signaling pathways, some clinical signs and symptoms of drug withdrawal were also introduced (Table 3).

## 5. Involvement of CaMKII and ERK in the DAergic System in the Nucleus Accumbens

Dopamine receptors are members of the G protein-coupled receptor superfamily. They are divided into two predominant groups, the D1-like (D1, D5) and D2-like (D2, D3, D4) receptors, based on the characteristic that activates adenylyl cyclase (AC) through binding with G_s/olf_ and inhibits AC through binding with G_i/o_ [116,117]. Almost 95% of neurons in the NAc are GABAergic MSNs [118]. The subpopulations of D1Rs and D1 MSNs in the NAc project to the substantia nigra and VTA via direct pathway, which leads to the basal ganglia nuclei that innervate non-basal ganglia areas, and project to the ventral pallidum (VP) via an indirect pathway, in which the nuclei innervate the basal ganglia areas [119]. D2 MSNs project to the VP through a direct and an indirect pathway, suggesting that some D1 MSNs and D2 MSNs have different projections [119]. Some studies have indicated that D1R and dopamine D2 receptor (D2R) are co-expressed in medium spiny projection neurons in the NAc and globus pallidus and form D1R-D2R heteromers in cell bodies and presynaptic terminals [120,121]. Our investigations showed that more than one-half of DA receptor-positive neurons are colocalized with both D1R and D2R in the mouse NAc.

D1R and D2R are both crucial in the context of drug abuse, including nicotine, cocaine, and other stimulant drugs. Passive inhalation of cigarette smoke significantly elevates the mRNA levels of both D1R and D2R in the rat NAc [122]. Subcutaneous administration of a mixed D1R and D2R antagonist *cis*-flupenthixol suppressed the escalation behavior in cocaine self-administration in rats [123]. D1 MSNs play an important role in learning acquisition based on the reward, and D2 MSNs are required for the switch when the learning strategy is altered [124]. CaMKII activity is elevated via Ca^2+^ release from the intracellular pool due to the stimulation of the D1 and D2 heteromer [125]. The stimulation of D1Rs in the NAc causes DA release, which activates cyclic adenosine monophosphate (cAMP) and protein kinase A (PKA), provoking cocaine-induced reinforcement and reinstatement [126,127]. Cocaine reinstatement behaviors induced by self-administration elevate CaMKII autophosphorylation and GluR1 phosphorylation levels in a D1R-mediated manner in the rat NAc shell and also increase the cell-surface expression level of GluR1-containing AMPA receptors in the NAc shell, suggesting that D1R activation and CaMKII are critical in drug addiction [60]. hM_4_D-CNO rats, animal models exhibiting elevated responses to methamphetamine relative to the vehicle, showed increased D1R and CaMKII expression levels in the dorsal striatum [128]. D1R mutant mice show impaired spatial memory acquisition and significantly decreased CaMKII and CREB phosphorylation levels relative to WT mice [129]. In addition, an epidemiologic study revealed that the DRD1 gene is correlated with nicotine addiction by investigating the relationship between five single-nucleotide polymorphisms within or near the DRD1 gene and nicotine addiction [130]. Administration of the D1R antagonist SCH-23390 blocked nicotine-induced single-pulse stimulation of DA release and locomotor sensitization in rats [131,132] and nicotine-induced dendritic remodeling in the NAc of adolescent rats [133]. The intracerebral infusion of the D1R antagonist SCH-39166 into the rat NAc shell could inhibit nicotine-induced CPP [134]. Taken together, D1R signaling in the NAc is essential for stimulant drug exposure through CaMKII activities.

D1R signaling also participates in the ERK signaling cascade. DA release is promoted by exposure to the stimulant drug, which contributes to the activation of D1R and increase of Ca^2+^ influx [87], and further activates PKA and Raf, which in turn activate Ras-Raf-MEK signaling and result in ERK phosphorylation. Some studies have reported that DARPP-32 is involved in D1R-mediated ERK phosphorylation in the mouse striatum, and DARPP-32 is activated by PKA and indirectly regulates ERK activity [135]. DARPP-32 mutant mice exhibit reduced ERK phosphorylation levels in the NAc and dorsal striatum [135]. Furthermore, the activation of D1R increases ERK phosphorylation induced by cocaine in the caudate-putamen [136]. The antagonism of D1R and NMDAR by pretreatment with SCH-23390 and MK801 significantly inhibits the phosphorylation of ERK in the NAc and dorsal striatum [88]. Systemic administration of SL327, an MEK inhibitor, prior to the administration of cocaine, blocks the cocaine-induced increase in ERK phosphorylation levels and hyperlocomotion [88]. Δ^9^-tetrahydrocannabinol-(THC) induces elevation of ERK phosphorylation levels in the NAc of rats, and the dorsal striatum is partially abrogated by MK801 and completely blocked by SCH-23390 treatment [137]. These observations indicate that the activation of D1R and NMDAR plays a critical role in the process of ERK phosphorylation.

D2R is divided into two spliced isoforms, the long isoform and the short isoform, which is based on the difference of a 29 amino acid insert in the third cytoplasmic loop [138]. Arginine-abundant domains of the N-terminal fraction in the third cytoplasmic loop of D2R are CaM-interacting sites. CaM, as a cellular Ca^2+^ sensor, plays a crucial role in activating ion channel-regulating enzymes that are involved in the cell cycle and development [139]. CaMKIIα interacts with D2R via the IL3 domain to regulate intracellular signaling, such as CaM-dependent signaling and ERK [140]. D2Rs are highly expressed in the striatum, NAc, and olfactory tubercle [117,141]; have an inhibitory function in regulating AC and calcium channels; and activate inhibitory G-protein-activated inwardly rectifying potassium channels (GIRK) [142]. D2Rs mediate various brain functions and modulate cognition, movement, and motivation, suggesting that D2Rs are crucial in the DAergic system and become pharmacological targets in the context of Parkinson’s disease, schizophrenia, and drug abuse [117,143]. D2R signaling contributes to opiate memory acquisition in the phases of chronic opiate exposure and withdrawal, while D1R signaling is necessary for acute opiate memory in the drug-naïve condition [144]. Significant Ca^2+^ signaling was detected in NG108-15 cells stably expressing D2L receptor following transfection relative to the D2S receptor [145]. An increase in intracellular Ca^2+^/CaMKII signals induced by D2R stimulation using quinpirole was observed in N108-15 cells overexpressing the D2L receptor [145]. Impaired locomotor activities and significantly decreased reinforcement to the reward effects of drug exposure, including ethanol and cocaine, were observed in D2R knockout (D2R^−/−^) mice [146,147,148]. Ca^2+^-dependent signaling is required for CaMKII activation, and ERK and BDNF are also altered in drug abuse through Ca^2+^ signaling stimulation [139,149]. Our previous investigations showed that chronic nicotine administration robustly elevates CaMKII and ERK phosphorylation levels as well as BDNF level in the WT mice in both the NAc and hippocampal CA1 region, whereas D2R^−/−^ mice show resistance in response to nicotine administration and failure of upregulation of above protein levels [56].

## 6. Involvement of LCPUFAs and the Novel Target FABP3 in Nicotine and Other Psychostimulant-Induced Addiction

Long-chain polyunsaturated fatty acids (LCPUFAs), which are abundant in the brain and retina, play an essential role in composing neuronal membrane phospholipids and developing the brain [150,151]. LCPUFAs including ω-3, such as docosahexaenoic acid (DHA) and eicosapentaenoic acid (EPA), and ω-6, such as arachidonic acid (AA), are not only involved in Alzheimer’s disease [152,153], schizophrenia [154] and other neurodegenerative diseases, but are also essential in psychostimulant-induced addiction. Rats receiving soybean oil, which is abundant with ω-6 fatty acids, exhibited significantly higher preference scores than the control group in amphetamine-induced CPP behavior, and showed anxiety-like behaviors in the elevated plus maze test [155]. A marked increase in protein carbonyl level was observed in the cortex and hippocampus in rats supplemented with food rich in ω-6 fatty acids, suggesting that uptake of ω-6 facilitates stimulant drug-induced addiction since protein carbonyl level is correlated with psychostimulant-induced preference and withdrawal behavior [155]. The endocannabinoid system, which regulates the pharmacological effects of cannabis, plays a role as a modification factor in the reinstatement effect of methamphetamine seeking behavior through the mediation of the AA cascade [156]. The administration of the selective D2R agonist quinpirole elevated AA metabolism and signaling in the vehicle group of rats, while these elevations were blocked by valproate treatment [157]. The results of analyzing regional brain incorporation coefficients following intravenous administration of radiolabeled AA suggest that D-amphetamine activates D2R signaling via the involvement of AA signaling [158]. However, the supplementation of fish oil, which is abundant in ω-3 fatty acid, led to a decreased preference score in amphetamine-induced CPP and the reduction of D1R and D2R expression levels [159]. Clinical investigations have demonstrated that six cocaine addicts with a history of aggressiveness show significantly decreased ω-3 fatty acid DHA in the measurement of plasma levels, suggesting that there is a possible correlation between ω-3 fatty acid deficiency and aggressiveness in humans [160]. These findings established that ω-3 LCPUFAs DHA, EPA, and ω-6 LCPUFA AA are involved in, have negative effects, and have positive effects in the context of psychostimulant-induced addiction, respectively.

Since LCPUFAs are insoluble in water, fatty acid-binding proteins (FABPs), which play the role of cellular shuttles, are necessary for intracellular trafficking [161]. FABPs interact with and as intracellular transporters of THC and cannabidiol in decreasing the metabolism of endocannabinoids [162]. Pretreatment with the FABP inhibitor SBFI26 significantly reduced ethanol consumption in mice as FABP5 and FABP7 inhibited the transport of anandamide to fatty acid amide hydrolase and elevated anandamide levels in the mouse brain, thereby increasing the preference and consumption of ethanol [163]. FABP 5/7 double knockout mice exhibit similar acquisition of cocaine-induced CPP relative to WT mice, whereas these mice do not exhibit stress-induced preference and show decreased levels of serum corticosterone under stress relative to WT mice [164]. Recent studies have reported that FABP5 knockdown in the adult rat NAc shell with RNA interference via adeno-associated viral vector attenuates cocaine self-administration and modulates the excitability of MSNs in the NAc shell [165]. FABP3 binds to ω-6 PUFAs [166], forms a complex, and colocalizes with D2R in the glutamatergic terminals and cholinergic interneurons in the dorsal striatum of mice [167]. FABP3^−/−^ mice show increased haloperidol-induced catalepsy, suggesting that FABP3 is involved in DAergic signaling [167]. In addition, FABP3^−/−^ mice exhibit cognitive dysfunction, hyperlocomotion, and impairment of fear extinction, which demonstrates that FABP3^−/−^ mice exhibit post-traumatic stress disorder (PTSD)-like behaviors [168].

The involvement of FABP3 in stimulant drugs, including nicotine-induced addiction, has not yet been fully investigated. Since there is no specific evidence suggesting that FABP3^−/−^ mice show D2R dysfunction and exhibit behaviors related to the DAergic system, we demonstrated that FABP3^−/−^ mice exhibit impaired CPP behaviors following chronic nicotine administration. This impairment was correlated with the lack of responsiveness of both CaMKII and ERK phosphorylation in the NAc. In FABP3^−/−^ mice, the number of D2R-positive neurons was obviously elevated, whereas the number of D1R-positive neurons and the responsiveness of c-Fos level in response to nicotine, which is closely correlated with CaMKII autophosphorylation levels, were significantly reduced, implying that FABP3^−/−^ mice show aberrant D2R signaling and further affect D1R and c-Fos signaling. These data suggest that FABP3 could be a novel target in nicotine-induced addiction and other drug abuses affecting DAergic signaling and provide evidence for developing novel therapies in the future. Intriguingly, the basal levels of CaMKII autophosphorylation and its downstream target are decreased in the D2R^−/−^ mice NAc and hippocampal CA1 region. However, significantly increased CaMKII autophosphorylation levels were observed in the NAc of FABP3^−/−^ mice. FABP3 is also a target in terms of PTSD-like symptoms as it is correlated with ω-3 and ω-6 PUFA transport [169], and ω-3 PUFA supplements ameliorate PTSD-like symptoms in patients [170]. FABP3^−/−^ mice show increased locomotor activities related to PTSD-like behaviors [168], whereas D2R^−/−^ mice exhibit hypolocomotion, implying a disturbance of DA receptor signaling [56]. Although an increase in intracellular Ca^2+^/CaMKII signaling induced by D2R stimulation is observed in NG108-15 cells overexpressing the D2L receptor, D2R partially but not completely colocalizes with FABP3 and mediates Ca^2+^/CaMKII signals independent of FABP3. FABP3 is possibly required to inhibit the function of D2R in D1R signaling since CaMKII autophosphorylation in D2R-positive neurons was increased in the NAc of FABP3^−/−^ mice. Consecutive nicotine administration produces DA level elevation and causes enhanced basal levels of cAMP/Ca^2+^ signaling through D1R activation, whereas this signaling negatively regulates D2R/FABP3 signaling. D1R activation provokes calcium influx, causing CaMKII activation and its migration to the nucleus, subsequently inducing enhancement of CREB/c-Fos signaling, which underlies the process of nicotine-induced addiction and other drug abuse. However, in FABP3^−/−^ mice, D2R/FABP3 signaling is impaired due to FABP3 deficiency, which eliminates the negative regulation of D2R/FABP3 signaling and leads to elevated cAMP/Ca^2+^ levels and CREB phosphorylation, as well as c-Fos activities. This constitutively increased signaling could not trigger nicotine addiction and possibly become the underlying mechanism of the failure of nicotine-induced addiction in FABP3^−/−^ mice (Figure 3).

## 7. Conclusions

This article reviewed the underlying brain mechanisms involved in nicotine-induced addiction and the fundamental function of CaMKII and mediated kinase ERK in nicotine-induced addiction and other drug abuse. The DA receptor-related signals in the NAc are crucial in the exposure to stimulant drugs, and perturbation of these signals causes impaired acquisition of psychostimulant-induced addiction. Finally, we introduced a novel protein target, FABP3, in nicotine-induced addiction, and elucidated the underlying mechanisms of the failed process in nicotine-induced addiction, demonstrating that FABP3 is an anticipated therapeutic target in nicotine addiction and other drugs of abuse affecting DAergic systems.

## Figures and Tables

**Figure 1 ijms-22-03189-f001:**
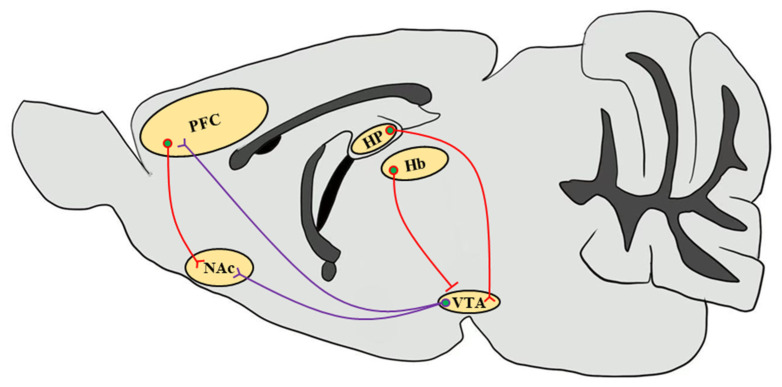
Simplified schematic representation of the mouse brain reward circuitry. The scheme of the mesolimbic DAergic projections (purple) of the mouse brain sagittal section emphasizes the predominant afferents that originate in the VTA and input into the NAc as well as the PFC, which are modulated by stimulant drug exposure through CaMKII functions. Glutamatergic inputs (red) that regulate neuronal circuits in response to drug abuse from the PFC to the NAc and from the HP to the VTA through the subiculum are highlighted [81]. Since CaMKIIβ regulates synaptic activity in the Hb in the context of depressive behaviors, it increases glutamatergic inputs onto VTA GABAergic interneurons to inhibit VTA DAergic neurons, thereby decreasing the DA release in the NAc [82,83]. VTA, ventral tegmental area; NAc, nucleus accumbens; PFC, prefrontal cortex; HP, hippocampus; Hb, habenula.

**Figure 2 ijms-22-03189-f002:**
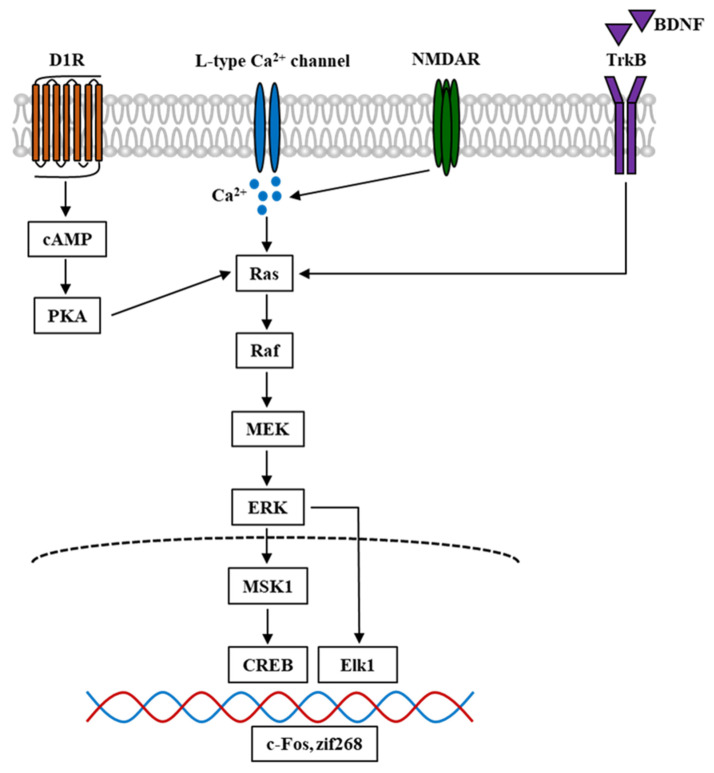
Simplified schematic diagram of ERK signal cascade in the mouse brain involved in psychostimulant-induced addiction. Nicotine and other stimulant drugs activate D1R and facilitate DA release in the DA terminals, leading to PKA and Raf activation. D1R activation causes L-type Ca^2+^ channels activation [92], contributing to Ca^2+^ influx and elevation of Ca^2+^ level, which leads to the activation of Raf. Raf is also activated by Ca^2+^ influx through NMDAR, which is mediated by the stimuli of Ras-GRF1 [93], and by BDNF through the binding and activation of TrkB. In turn, Raf activation leads to the phosphorylation and activation of MEK and ERK. The phosphorylation of ERK evokes the activation of downstream targets such as Elk1 and MSK1. The latter could activate and phosphorylate CREB. These transcriptional factors cause the transcription of IGEs such as c-Fos and zif268, which become the underlying mechanism of psychostimulant-induced addiction. D1R, dopamine D1 receptor; DA, dopamine; PKA, protein kinase A; NMDAR, N-methyl-D-aspartate receptor; Ras-GRF1, Ras protein-specific guanine-nucleotide releasing factor; BDNF, brain-derived neurotrophic factor; TrkB, tropomyosin receptor kinase B; MEK, MAPK-ERK kinase; MSK1, mitogen- and stress-stimulated kinase 1; CREB, cAMP response element-binding protein; IGEs, immediate early genes.

**Figure 3 ijms-22-03189-f003:**
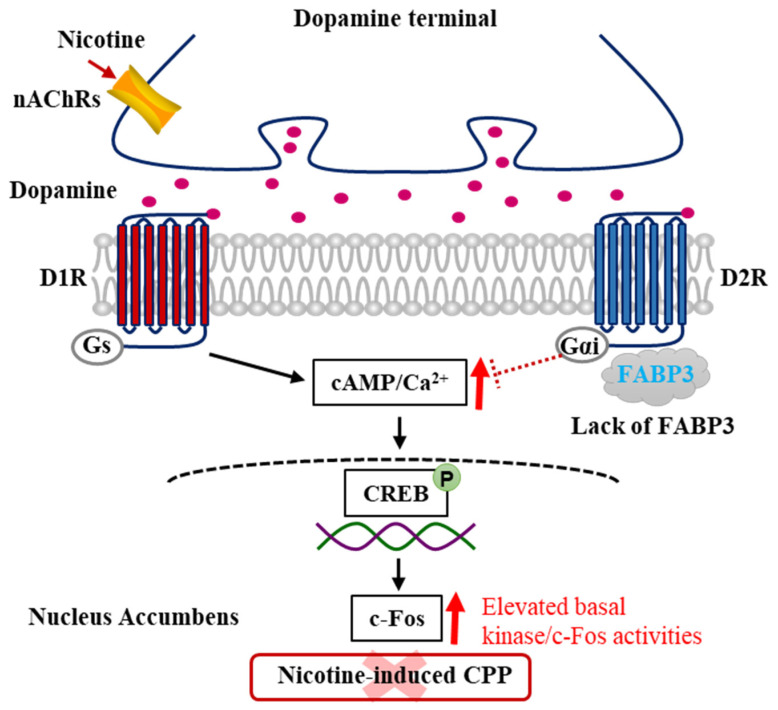
Schematic model of dopamine signaling pathways underlying nicotine-induced addiction in the FABP3^−/−^ mice NAc. In FABP3^−/−^ mice, the lowered cAMP/Ca^2+^ levels regulated by D2R/FABP3 signaling derepresses due to the impaired D2R function through the lacking FABP3, which provokes elevated cAMP/Ca^2+^ levels and increases CREB/c-Fos signaling. These constitutive elevated cAMP/Ca^2+^ levels and signals underlie the mechanism of failed acquisition of nicotine-induced addiction in FABP3^−/−^ mice.

**Table 1 ijms-22-03189-t001:** Evidence of CaMKII activity fully involved in the brain regions of circuitry with various drugs of abuse.

Drug	Region	CaMKII Activity
Nicotine	Nucleus accumbens	Phosphorylation level increased [56]
	Hippocampal CA1	Phosphorylation level increased [56]
	Ventral tegmental area	Activity increased [57]
Morphine	Nucleus accumbens shell	Activity is crucial for morphine-seeking [71]
	Hippocampus	CaMKIIα level increased in synaptosomes [72]
	Prefrontal cortex	Phosphorylation level increased [73]
Cocaine	Nucleus accumbens shell	Activity is crucial for cocaine-seeking [65,66]
	Hippocampal dentate gyrus	CaMKIIα phosphorylation mediates neuronal activation [72]
	Medial prefrontal cortex	CaMKIIα phosphorylation level increased after cocaine withdrawal [69]
Amphetamine	Nucleus accumbens shell	Activity is crucial for amphetamine-induced DA release [74]
	Hippocampus	Activity is crucial for amphetamine-induced CPP [75]
	Striatum	Phosphorylation level increased [76]

**Table 2 ijms-22-03189-t002:** Evidence that ERK activity is fully involved in the circuitry of brain regions for various drugs of abuse.

Drug	Region	ERK Activity
Nicotine	Nucleus accumbens	Phosphorylation level increased [56]
	Prefrontal cortex	Phosphorylation level increased [105,106]
	Hippocampal CA1	Phosphorylation level increased [56]
Morphine	Nucleus accumbens	Phosphorylation level increased [105]
	Prefrontal cortex	Phosphorylation level increased [105]
	Ventral tegmental area	Phosphorylation level increased [107]
Cocaine	Nucleus accumbens	Phosphorylation level increased [105]
	Prefrontal cortex	Phosphorylation level increased [10,105]
	Striatum	Activity increased [88]
Amphetamine	Nucleus accumbens	Activity is crucial for amphetamine-induced CPP [108]
	Medial prefrontal cortex	Phosphorylation level increased at synaptic sites [109]
	Striatum	Phosphorylation level increased [76,110]

**Table 3 ijms-22-03189-t003:** Clinical signs and symptoms as a consequence of various drugs of abuse in activation of CaMKII and ERK signaling pathways.

Drug	Clinical Signs of Drug Withdrawal
Nicotine	Anxiety, Depression, Craving, Restlessness and impatience, Increased appetite and body weight [111]
Morphine	Anxiety, Nausea, Emesis, Diarrhea, Body aches, Restlessness, Agitation and dysphoria [112,113]
Cocaine	Anxiety, Depression, Craving, Restlessness, Vivid, unpleasant dreams, Fatigue and exhaustion, Increased appetite [114]
Amphetamine	Anxiety, Depression, Irritability, Body aches, Impaired social functioning Vivid, unpleasant dreams, Fatigue, Increased appetite [115]

## Data Availability

The data that support the findings of this study are available from the corresponding author upon reasonable request.

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
