# Peer review of "The Role of CaMKII and ERK Signaling in Addiction"

_ijms, 2021, doi:10.3390/ijms22063189_

Round 1

Reviewer 1 Report

Comments to the Author
The report by Jia et al. is an important review focusing on the “role of CaMKII and ERK signaling in addiction”. This review shows the molecular mechanisms and crucial roles of CaMKII and extracellular signal-regulated kinase (ERK) in drug-induced addiction. Additionally, the correlation between dopamine (DA) in nicotine-induced addiction is described. In my opinion, the article is transparent and contains sufficient descriptions of important signaling patways in correlation with pathology. 

Please find a minor concern below:
It would be important to include a summary table of clinical signs as a consequence of 
nicotine, morphine, cocaine and amphetamine in activation of CaMKII and ERK signaling pathway. 

Author Response

Reviewer #1:

The report by Jia et al. is an important review focusing on the “role of CaMKII and ERK signaling in addiction”. This review shows the molecular mechanisms and crucial roles of CaMKII and extracellular signal-regulated kinase (ERK) in drug-induced addiction. Additionally, the correlation between dopamine (DA) in nicotine-induced addiction is described. In my opinion, the article is transparent and contains sufficient descriptions of important signaling pathways in correlation with pathology.

Please find a minor concern below:

It would be important to include a summary table of clinical signs as a consequence of

nicotine, morphine, cocaine and amphetamine in activation of CaMKII and ERK signaling pathway.

Ans: According to the comment, we added a table of clinical signs and symptoms as a consequence of various drugs of abuse in activation of CaMKII and ERK signaling pathway as Table 3.

Reviewer 2 Report

The authors reviewed the basic brain mechanisms involved in nicotine addiction and the fundamental functions of CaMKII and nicotine-mediated ERK in nicotine addiction and other drug abuse. The authors also discussed dopamine receptor (DA) signaling involved in 16 nicotine-induced addictions in the brain's reward circuitry. In addition, the authors elucidated the mechanisms underlying the failed process in nicotine addiction, demonstrating that FABP3 is the predicted therapeutic target for addiction to nicotine and other overused drugs affecting DAergic systems.

The literature review was done very carefully. I think that it fully exhausts the issues raised by the authors. I enjoyed reading this paper.

Author Response

Reviewer #2:

The authors reviewed the basic brain mechanisms involved in nicotine addiction and the fundamental functions of CaMKII and nicotine-mediated ERK in nicotine addiction and other drug abuse. The authors also discussed dopamine receptor (DA) signaling involved in 16 nicotine-induced addictions in the brain's reward circuitry. In addition, the authors elucidated the mechanisms underlying the failed process in nicotine addiction, demonstrating that FABP3 is the predicted therapeutic target for addiction to nicotine and other overused drugs affecting DAergic systems.

The literature review was done very carefully. I think that it fully exhausts the issues raised by the authors. I enjoyed reading this paper.

Ans: Thank you for your encouragement for our manuscript.